# The Current Quality of Web-Based Information on the Treatment of Bipolar Disorder: A Systematic Search

**DOI:** 10.3390/jcm11185427

**Published:** 2022-09-15

**Authors:** Martina Piras, Alessandra Perra, Oye Gureje, Antonio Preti, Mauro Giovanni Carta

**Affiliations:** 1Innovation Sciences and Technologies, University of Cagliari, 09124 Cagliari, Italy; 2Department of Medical Sciences and Public Health, University of Cagliari, 09124 Cagliari, Italy; 3Department of Psychiatry, University College Hospital, Ibadan 200285, Nigeria; 4Department of Neuroscience, University of Turin, 10126 Turin, Italy

**Keywords:** bipolar disorder, treatment, information dissemination, quality, health literacy

## Abstract

**Background:** An important aspect of managing chronic disorders like bipolar disorder is to have access to relevant health information. This study investigates and compares the quality of information on the treatments of bipolar disorder that is available on English websites, as an international language, and on Italian websites, as a popular local language. **Methods:** A systematic review search was obtained from four search engines. We excluded unrelated materials, scientific papers, and duplicates. We analyzed popularity with PageRank; technological quality with Nibbler; readability with the Flesh Reading Ease test and Gulpease index; quality of information with the DISCERN scale, the JAMA benchmark criteria, and on the extent of adherence to the HONCode. **Results:** 35 English and 31 Italian websites were included. The English websites were found to have a higher level of quality information and technological quality than the Italian ones. Overall, the websites were found to be difficult to read, requiring a high level of education. **Conclusions:** These results can be important to inform guidelines for the improvement of health information and help users to reach a higher level of evidence on the websites. Users should find the benefits of treatment, support for shared decision-making, the sources used, the medical editor’s supervision, and the risk of postponing treatment.

## 1. Introduction

Bipolar disorder (BD) is a multicomponent disorder characterized by severe mood disturbance, neuropsychological deficits, immunological and physiological changes that negatively impact the lives of affected persons [1,2]. Affecting about 1% of the population in its most severe forms and as many as 3% in less severe forms, it is a serious public health problem globally [3]. In Italy, the prevalence of BD in the general population is estimated to be about 2%, rising to 6% when considering the broad bipolar spectrum [4]. As elsewhere in the world [5], a lower fraction of people living with BD in Italy is in current treatment than the estimated prevalence in the general population. For example, the estimated treated annual prevalence rate of diagnosed BD was 1 per thousand in Lombardy, Northern Italy [6]. A scant minority of those treated in community settings (0.7% to 6.1% depending on treatment) receive psychosocial treatments such as individual psychotherapy, couple/family therapy, group psychotherapy, and family interventions [6]. Psychosocial treatments may improve the course of BD [7]. However, according to current guidelines, pharmacotherapy, in particular with mood stabilizers, is the main therapeutic indication in the treatment of BD for both acute episodes and relapse prevention [8,9,10].

BD is one of the leading causes of disability worldwide [11,12,13,14] and is also associated with high rates of premature mortality from both suicide and medical comorbidities [15,16], thus making the condition a serious public health problem and one with a high level of considerable negative social costs [17]. Persons living with this disorder often have low levels of insight and may have poor treatment adherence [18,19], which may constitute a barrier to searching for and obtaining evidence-based information that may be relevant to their health and wellbeing [20,21]. Another consistent barrier that discourages help and information seeking is the stigma of BD [20,22,23].

Indeed, stigma in mental health is an important barrier to care [24]. Fear related to stigma acts as a sociocultural barrier to treatment [25], as does low confidence in professional healthcare services [26]. It is long known that patients with a mental disorder may refrain from seeking help because of the perceived stigma surrounding their condition [27,28]; more generally perceived public stigma, especially when associated with internalized stigma, contributes to a delay in seeking medical help [28,29]. Conversely, knowledge of help-seeking options and available treatments affects the initiation of treatment after the onset of a mental disorder [30].

People living with BD often experience public stigma via newspapers, films/TV series, and celebrity self-disclosure [31]. In patients with BD, perceived stigma is as high as in people with schizophrenia [32], and is associated with worse medication adherence [32].

Because of its impersonal nature, ease of use, and anonymity, the use of the Internet may be favored in patients with higher perceived public stigma. Indeed, when the topic was investigated, it was found that people living with some stigmatized illness were significantly more likely to have used the Internet for health information and to have increased utilization of health care based on information found on the Internet than those with non-stigmatized conditions [33]. This is one possible reason for people with severe mental disorders, such as schizophrenia and chronic depression, to use the Internet as a source of health information more than the general population [34]. Conversely, the Internet may become the target of campaigns aimed at reducing the stigma associated with mental disorders and increasing willingness to access care [28,35].

Information seeking is an important aspect of managing and coping with chronic disorders [36,37], and people living with BD have often expressed the need to obtain information about their illness [38,39,40,41]. Even though people living BD may prefer to get such information through face-to-face conversations with their physicians, as do patients with other mental and physical illness [40,42,43,44], this may not always be feasible due to the limited time available for direct access to the physicians [41].

The internet is increasingly used as a source of information on health issues [45,46] and has become particularly so for mental health [47,48]. For example, about 59% of the U.S. population and 74% of college students search for health information on the Internet [49,50]. A survey involving about one thousand patients with bipolar disorder in over 17 countries found that 77% of them have sought information about their condition on the Internet [38]. Indeed, both patients and health providers consider the internet as an important source of medical information, including information about bipolar disorder. Among the features that make the Internet useful for this purpose there appear its global availability, access at the point or time of need by the individual, and the fact that it provides for anonymity, a feature that may be particularly important for persons seeking information about mental health issues [38,51,52,53].

While providing high quality internet information may help people to make informed choices about their treatment, previous studies have found that the quality of available information on mental health is relatively poor [54,55,56,57,58,59].

In the field of BD, several studies in the past have analyzed whether in people with BD the experience of seeking online health information had been useful and reliable [20,38,41,60]. According to some of these studies, persons living with BD and their families reported that finding relevant information is often a challenge because several guidelines provide conflicting information and the message is often couched in technical and complex terminologies [17,43,61]. To date, two studies have evaluated the quality of online health information in websites specifically [48,50]. The first one analyzed two search engines only, evaluated their quality and readability, and showed some statistical limits, especially with the DISCERN scale, in order to predict the quality of online health information [48]. The other study compared the quality of information about different mental health disorders and showed that online BD health information is better than other mental health information that can be retrieved online [50]. However, online health information evolves and changes rapidly [62]. For this reason, it is important to analyze the existing websites dedicated to online BD health information sometime after the previous studies about BD, and use a multidimensional method of evaluation that includes different quality information instruments (in order to avoid the statistical limit of some scale), readability index, and also technological measures (like usability and accessibility). Readability, usability, and accessibility are key features in making information on websites truly accessible to users. So far, no study has evaluated the language differences in online BD information in order to understand possible language barriers that prevent achieving a high level of health information online. Since BD benefits from multi-professional treatment, including pharmacotherapy, psychotherapy, and additional psychosocial and rehabilitation interventions [8,9,10], it is expected that websites provide information about the different treatments (pharmacotherapy, psychotherapy, psychoeducation program) available in different languages according to current guidelines, and that they comply with information quality criteria [63].

## 2. Aims

The aim of this study is to analyze the readability, accessibility, technological usability, and the quality of the information on websites dedicated to the dissemination of information on the treatment of bipolar disorder and on currently available interventions. Taking cognizance of the extensive use of the Internet all over the world, we decided to compare websites in two different languages: English, considered as an international language of scientific dissemination, and Italian, which is our main local language.

## 3. Methods

### 3.1. Search Strategy

A systematic review in the form of online search was conducted using the three most common search engines “Google” (www.google.com, accessed on 12 April 2021), “Bing” (www.bing.com, accessed on 12 April 2021), “Yahoo” (www.yahoo.com, accessed on 12 April 2021), and one independent search engine “DuckDuckGo” (duckduckgo.com/, accessed on 12 April 2021), which aims at preserving the privacy of the users. These search engines host over 98% of all searches worldwide and across different platforms, such as desktop, tablet, or mobile [64]. Overall, there is evidence that 80% of searches on the Internet for health-related information are performed with a search engine, both by the general public and among those diagnosed with bipolar disorder [38]. Thus, our strategy is likely to intercept the majority of the websites that are accessed by people looking for information about bipolar disorder treatment.

Two key terms used were: “Bipolar disorder treatment” for search in English and “Trattamento disturbo bipolare” for search in Italian. The websites have been selected in order of appearance, and the first 20 results in each search engine and for each language were included. This method was chosen because there is evidence that users concentrate their exploration of the websites that are retrieved from a search engine to the first ten entries and rarely go beyond the first two pages of the results [1,60]. It was decided to exclude discussion or forums websites, websites requiring password or payment, non-written documents such as videos, only-title text, advertisements and scientific papers. The websites were assessed from 12–20 April 2021. The authors applied the flow diagram PRISMA [65] adapted for online engine search.

### 3.2. Assessment

The evaluation was carried out independently by two researchers, who then had their results harmonized. The popularity of the website was checked via Google’s PageRank with https://checkpagerank.net/, accessed on 20 April 2021 (Google’s page rank)—Google’s PageRank is one of the methods Google uses to determine a page’s relevance or importance [66]. 

The technological quality of the website was checked with Nibbler at https://nibbler.silktide.com/, accessed on 20 April 2021 including the following indexes: overall, accessibility; experience; marketing, technology, and mobile. Each website was assessed for its accessibility (such as ease of locating information on the website, URL format, and page titles), the rated user experience (such as the content value, format, mobile availability, internal links, etc.), marketing (links to social media, popularity, meta tags, freshness, etc.), and the quality of IT used [57,67].

As for information in the English language, websites readability was assessed using the Flesh Kincaid Reading Ease and the Flesh Kincaid Grade Level, tested with the readability test tool of WebFX at https://www.webfx.com/tools/read-able/, accessed on 20 April 2021. The Flesch Reading Ease score takes into account factors, such as the number of words per sentence and the number of syllables per word, to generate a score from 0 to 100, with a high-scoring text being more easily understood than one with a low score. A text with a score of 71–100 is considered ‘easy’ to read, with an average 11-year-old able to read it with ease. A score of 61–70 is considered of ‘standard’ difficulty, with children aged 13–15 years being able to read it. A text with a score of 60 or below is considered ‘difficult’ to read [52,68].

The readability of the Italian websites was assessed using the Gulpease readability Index [69], testing at the following address https://farfallaproject.org/readability_static/, accessed on 20 April 2021. The Gulpease index takes into account the length of a word in characters rather than in syllables, which proved to be more reliable when assessing the readability of Italian texts. The index ranges from 0—that means lowest readability—to 100, maximum readability [70]. Since the Flesh Kincaid reading test and the Gulpease readability Index, which is tailored for the Italian language, are not directly comparable, readability has been compared between languages on the degree of complexity. The degree of complexity has been grouped into three classes: “easily readable” (for texts that the average 11-year-old should be able to read); “standard level of read- ability” (for texts that children aged 13–15 years old should be able to read); and “difficult to read” (for texts that require a high-school level of literacy or higher). 

Quality of the information provided by the website, on the specific web page dedicated to the topic of interest, was assessed with the DISCERN scale, the JAMA benchmark criteria, and adherence to Health on the Net code (HONcode). DISCERN is a tool designed to help users consult health information to judge the quality of written information provided on treatment choices [71]. It consists of 16 items and each criterion is rated on a scale from 1 to 5. The higher the level, the better the quality of information. DISCERN is divided into three main sections in assessing reliability: whether it can be trusted as a source of information about treatment choice; the quality of information; and the overall quality [52,72]. As in past studies, we also calculated a global DISCERN score as a sum of all items, and following Khazaal et al. (2012) recommendation, we used a DISCERN score cutoff of ≥40 to identify the websites with enough adequate information quality.

The JAMA benchmark criteria range from 0 to 4, and it is aimed at critically assessing the credibility and utility of medical information read on the Internet [73]. The JAMA benchmark criteria assess the following core standards: web-site authorship had to formally include authors, contributors, affiliations, and credentials; attribution should include references and sources used for the content and copyright information; disclosures should include details about sponsorship, advertising, commercial funding, potential conflicts of interests; currency should include the date of posted information and its update [73]. The HONcode certification was proposed by the Health On the Net Foundation (HON) and certifies the quality of the medical information provided on the Internet [74].

## 4. Statistics

All data were coded in Excel and analyzed using the Statistical Package for Social Sciences (SPSS) version 20. Non-parametric statistics were used to compare indicators of interest between English and Italian websites. The intra-rater reliability of the scale for the JAMA benchmark criteria and for the DISCERN scale was measured with the intraclass correlation coefficient (ICC), with a 95% Confidence Interval (CI). ICC values ≥ 0.60 are considered acceptable [75].

## 5. Results

The initial sample consisted of 160 websites (80 Italian websites and 80 English websites). We excluded 36 Italian duplicate websites and 27 English duplicate websites. At the end of the screening, we excluded 13 Italian websites (3 scientific papers, 9 documents or non-written information, and 1 was not pertinent) and 18 English websites (12 scientific papers, 5 documents or non-written information, and 1 video information). Finally, we analyzed 31 Italian websites and 35 English websites (Figure 1).

For both languages, the most numerous sites were elected by the search engine Google (Ita. = 16, Eng. = 17), followed by Yahoo 19 (Ita. = 11, Eng. = 8); from the search engine DuckDuckGo only sites in English were elected—9, while Bing included websites from both languages 5 (Ita. = 4, Eng. = 1) (Table 1).

### 5.1. Google PageRank Score

As expected, the Google PageRank score was higher for the English websites (6.4 ± 2.0) than for the Italian ones (3.6 ± 1.6), as a reflection of the Anglo-centric nature of this measure. Overall, the PageRank score of the websites about the treatment of bipolar disorder was not good, since a score of 10 is expected for the most authoritative websites.

### 5.2. Technological Quality of the Websites

The technological quality of the websites concerning the treatment of bipolar disorder was higher in English than Italian websites. The Global average Nibbler score was 88.7% in English websites and 85.5% in Italian websites (Mann-Whitney U = 324.50; z = −2.80; *p* = 0.005). The difference was principally attributable to Experience (86.2% versus 80.7%; Mann-Whitney U = 324.00; z = −2.81; *p* = 0.005) and Marketing (84.2% vs. 67.9%; Mann-Whitney U = 222.50; z = −4.11; *p* < 0.0001). No statistically significant differences were found in Accessibility (91.6% vs. 93.1%; Mann-Whitney U = 536.00; z = −0.84; *p* = 0.933), Technology (89.8% vs. 88.7%; Mann-Whitney U = 439.00; z = −1.33; *p* = 0.182), and suitability for Mobile phones (94.4% vs. 88.4%; Mann-Whitney U = 513.50; z = −0.52; *p* = 0.602).

### 5.3. Readability

Readability, as measured with the Flesh Kincaid test, suggested that most English websites required reading skills at high school level or higher (85.7% of cases). Italian websites, as well, required high reading skills at high school levels (90.3% of cases). Overall, websites reporting information about the treatment of bipolar disorder were not too easy to read.

### 5.4. Adherence to the HONCode

English websites were hugely more likely to report adherence to the HONCode than Italian websites: n = 14 (40%) versus n = 4 (13%); χ^2^_Yates_ = 4.79; df = 1; *p* = 0.029. Overall, less than a third of the websites were compliant with the HONCode adherence.

### 5.5. Information Quality According to the JAMA Benchmark Criteria

The Intra-rater reliability for the JAMA benchmark criteria was acceptable for the English websites (ICC = 0.674; CI = 0.453–0.820). However, it was poor for the Italian websites (ICC = 0.320; CI = 0.0–0.640). Thus, results concerning the JAMA benchmark criteria should be considered with caution. Requirements for Attribution were less often respected in English than Italian websites χ^2^ = 10.54; df = 4; *p* = 0.032). However, English websites were more likely than Italian websites to respect the requirements of Disclosure χ^2^ = 17.05; df = 1; *p* = 0.002). There were no differences as far as Authorship and Currency were concerned (Figure 2).

### 5.6. Information Quality According to the DISCERN Scale

Intra-rater reliability for the DISCERN scale was excellent in both the English (ICC = 0.915; 0.867–0.951) and the Italian (ICC = 0.874; CI = 0.798–0.930) websites.

English websites were rather poor at reporting the impact of treatment on quality of life, at describing how the treatments work, and what happens if someone does not undertake the proposed treatment (Figure 3). The same limitations were observed for Italian websites, with an additional poor reporting of indications about shared decision-making (Figure 4). Overall, there were no statistically significant differences between English and Italian websites as far as Reliability and overall Quality of information are concerned, according to the DISCERN scale (Mann-Whitney U: both comparisons, *p* > 0.30). The DISCERN global score also did not differ between the English (51.5 ± 10.4) and the Italian websites (49.6 ± 9.3): Mann-Whitney U = 472.00; z = −0.91; *p* = 0.365. Around 80% of the websites reached the threshold for acceptable information quality on the DISCERN scale, again with no difference between the English (80%) and the Italian (84%) websites: χ^2^_Yates_ = 0.01; df = 1; *p* = 0.931.

### 5.7. Information Quality According to BD Treatment Guidelines

The English websites explained better and with more details the information related to different treatments of proven effectiveness (Table 2). The Italian websites rarely cited psychosocial interventions and focused more on pharmacological treatment.

Compared to the Italian ones, the English websites were more likely to provide a multidimensional view of the available treatments, with a greater attention to psychosocial and rehabilitation interventions.

## 6. Discussion

This study highlights what kind of information is present on Internet websites dedicated to the promotion of information for the treatment of bipolar disorder for online users. To better understand this topic, we assessed the quality of the information on different websites in a language used for international dissemination, English, and a popular local language, Italian, and we focused on the level of quality and how differences in quality may affect the information objectives. As expected and seen in other studies [57,67], the websites in Italian were found to be more difficult to read than the websites in English, although a high level of education was required for the websites in both languages. The websites in English had a higher level of technological profile, mainly regarding the user experience and marketing. However, both the websites in English and those in Italian had a good level of accessibility, technology and visualization on mobile devices, a key current feature, as much of the web traffic now go to cell phones. The analysis based on the JAMA Benchmark criteria showed acceptable levels for the English websites, while they were poor for the Italian ones: in particular, the Italian websites respected less the Disclosure requirements, while the Attribution requirements were lower in the sites in English, with no relevant differences seen for Authorship and Currency. Adherence to the HONCode was much higher in the English websites than in the Italian ones (40% vs. 4%), even though overall, less than a third of the websites adhered to the HONCode. It should be noted that the proportion of English websites with adherence to the HONCode in this study (40%) was comparable to the results of a past study on the topic (44%) [48]. A lower proportion of adherence to the HONcode (28.6%) was found in a more general study aimed at investigating the quality of mental health information on the Internet [50]. In a previous study, the DISCERN score and readability did not predict the quality of the retrieved websites [48]. In the present study, the DISCERN scale indicated that the websites in English were poor in reporting the impact of treatment on quality of life and in describing how treatments work and what happens if someone does not undergo the proposed treatment. As for the websites in Italian, the DISCERN scale highlighted the same limitations, with a further limitation related to indications for the shared decision-making process. Overall, no differences emerged on the DISCERN scale between the English and Italian websites as regards the reliability and overall quality of the information. Despite the limitations of the information quality of the surveyed websites, the overall quality of these websites, as assessed with the global score of the DISCERN, was acceptable in about 80% of cases. The JAMA Benchmark criteria also showed acceptable levels for the English websites, but not for the Italian ones. However, when applying more stringent criteria for the evaluation of the quality of information describing the treatment of bipolar disorder, currently available websites have several shortcomings that need to be addressed. This is especially important since there is evidence that the digital health literacy of patients with bipolar disorder is sometimes modest, and they may require some support to safely navigate web-based health resources [20]. 

In general, and considering the impact of BD, it is important that people living with the condition, as well as the general population, have access to high-quality Internet information. Moreover, Internet users need to be able to find information about major guidelines, including those providing information on different treatments, lifestyle changes [2] and how to have access to professional support for decision-making [57,76]. These represent the major point to be improved in the BD online websites. In particular, the availability of different treatments and especially of psychosocial programs is often not clearly specified, and it is less often described on Italian websites. In summary, there are quality differences depending on the language of the websites, in particular, and this is associated with differences in the description of available treatments according to current guidelines on BD [8,9,10]. A better description of the available treatments, including pharmacotherapy, psychotherapy, and additional psychosocial and rehabilitation interventions, may help users to better understand and plan their perspective of treatment, favoring an informed choice. 

### Implications for the Clinics

This study could have important implications for the clinics. Indeed, having access to effective intervention information is a goal in managing the disorder. For this reason, understanding and evaluating the quality of online health information through a systematic review search is the first step to filling the gap between guidelines and real online information in order to promote and raise the awareness of the different stakeholders (online users and websites developers) to search and create good quality health information. Good quality health information, compared to the actual health information to be found online, needs to be improved in the field of the explanation of different treatments. There are differences also in the access to health information, depending on the language. Achieving this quality standard for all languages could better support people who live with BD in managing the disease and in accessing clear information that can direct these people to effective help seeking. 

## 7. Conclusions

The quality of information on the Internet does not reach a high level worldwide yet. In the present-day digital era, the evaluation and monitoring of the standard of health information on websites is a very important area in need of investigation. This is with a view to guaranteeing all people equal rights to high-quality access to health information, especially to information about mental health, given the pervasive stigma that is still associated with mental health conditions. This study is an attempt to contribute to meeting such a need.

## 8. Strengths and Limitations

The findings of the study have to be considered within its strengths and limitations. One important strength is that the method used reflects a multidimensional approach to defining the quality of Internet information. However, given that our analysis is based on two major languages, more studies from a variety of language groups may be required to provide a broader international perspective.

## Figures and Tables

**Figure 1 jcm-11-05427-f001:**
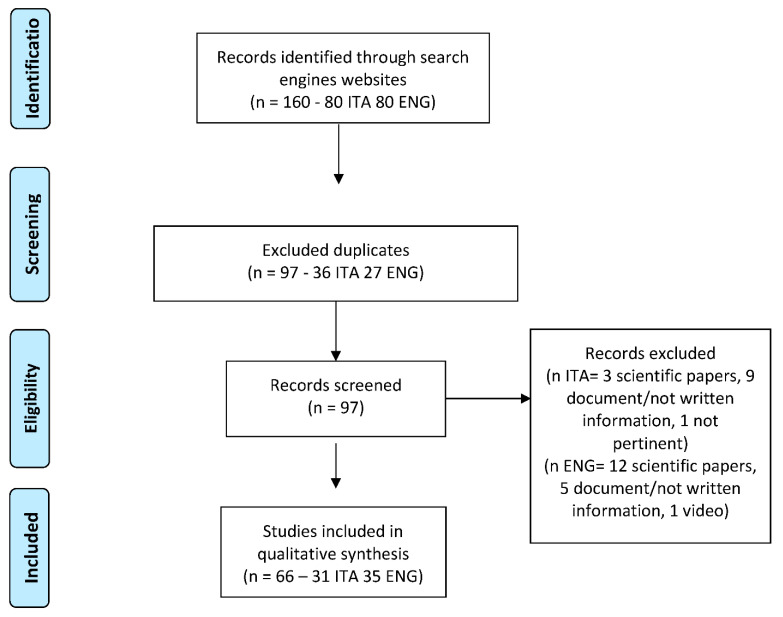
PRISMA Flow diagram of search results.

**Figure 2 jcm-11-05427-f002:**
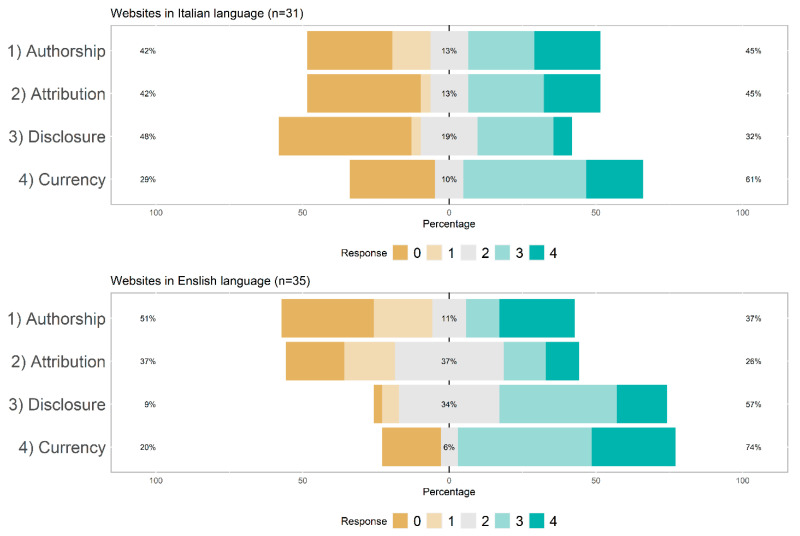
Distribution of the adherence to the JAMA benchmark criteria in Italian and English websites.

**Figure 3 jcm-11-05427-f003:**
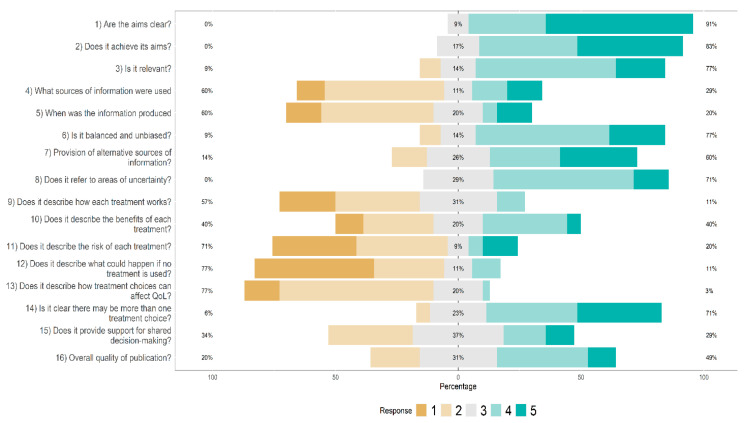
Distribution of DISCERN-scores combined and averaged across English websites (1 = low, 5 = high). Adapted from Charnock et al. [72].

**Figure 4 jcm-11-05427-f004:**
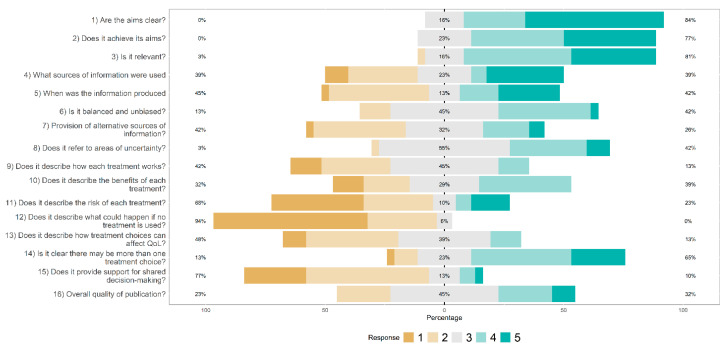
Distribution of DISCERN-scores combined and averaged across Italian websites (1 = low, 5 = high). Adapted from Charnock et al. [72].

**Table 1 jcm-11-05427-t001:** Distribution of elected web sites by search engine.

		Italian (n = 31)	English (n = 35)	Tot (n = 66)
**Search engine**	Google	16	17	33
	Yahoo	11	8	19
	Bing	4	1	5
	DuckDuckGo	0	9	9

**Table 2 jcm-11-05427-t002:** Distribution of the treatments for BD as reported in the Italian and English websites.

	English Websites (n = 31)	Italian Websites (n = 35)	Statistics
Drugs	27 (87%)	28 (80%)	χ^2^ = 0.59; df = 1; *p* = 0.44
Psychotherapy	25 (80%)	23 (67%)	χ^2^ = 1.85; df = 1; *p* = 0.17
Psychosocial Interventions	23 (73%)	9 (27%)	χ^2^ = 15.5; df = 1; *p* < 0.0001
Only one treatment	6 (20%)	16 (47%)	χ^2^ = 5.14; df = 1; *p* = 0.02
Two treatments	6 (20%)	12 (33%)	χ^2^ = 1.85; df = 1; *p* = 0.17
All types of interventions	19 (60%)	7 (20%)	χ^2^ = 11.7; df = 1; *p* < 0.001

## Data Availability

The datasets could be available with author request.

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
