# Peer review of "The Current Quality of Web-Based Information on the Treatment of Bipolar Disorder: A Systematic Search"

_jcm, 2022, doi:10.3390/jcm11185427_

Round 1
Reviewer 1 Report
The paper by Piras et al. provides a review of the current English and Italian websites focused on the treatment of bipolar disorder. This is a very interesting approach that raises important points, since the access to information on mental health is at the same time crucial and challenging, due to possible lack of quality and ethical issues. The paper is well written, the aims are clear, the results are adequately discussed. I would suggest to address minor points:
- In the Introduction, Authors should briefly discuss the issue of stigma in mental health, since this may represent one of the main barriers to adequate information.
- Previous studies on the topic of web information in BD are cited by the Authors, but more space should be dedicated to them in the Introduction, also to clarify why a review with this specific focus was needed.
- Authors compare English and Italian websites since the group is based in Italy; further information on the prevalence and clinical impact of bipolar disorder in Italy should be specifically provided at the beginning of the paper.
- I believe that describing the paper as a “systematic review” is not completely appropriate, since systematic reviews are usually meant to follow specific guidelines (e.g., PRISMA) concerning the search strategy and the paper presentation. The term “systematic review” may sound confusing in this case.
- In the Discussion section, I would suggest to add some sentences concerning the possible clinical implications of the results from this search. How would these be useful for improving the clinical management of bipolar disorders?
- The Limitations section should be expanded a bit. Indeed, there are some further points that could be addressed, e.g., some of the inclusion criteria are merely operational, such as the choice to include only the first 20 websites in each search engine.
- There is something wrong with references cited along the text since they do not appear in consecutive order.
- Punctuation and spaces should be revised. English language in some sentences should be revised as well.
Author Response
Reviewer #1:
Thank you very much for your kind consideration of our study.
- In the Introduction, Authors should briefly discuss the issue of stigma in mental health, since this may represent one of the main barriers to adequate information.
- Thank you for this comment, which prompted us to integrate the manuscript with a better presentation of the background and integrate the multidimensional variables that promote online health information search. In the introduction we have added a paragraph with references about the stigma barrier on Bipolar disorder.
- Previous studies on the topic of web information in BD are cited by the Authors, but more space should be dedicated to them in the Introduction, also to clarify why a review with this specific focus was needed
- Thank you for the suggestion. We improved the closing part of the introduction with information about other studies on mental disorders, and on the specific field of Bipolar Disorder. We also added some considerations about previous studies that led us to conduct this multidimensional method evaluation of specific health websites discussing about BD.
- Authors compare English and Italian websites since the group is based in Italy; further information on the prevalence and clinical impact of bipolar disorder in Italy should be specifically provided at the beginning of the paper.
- Thank you for the comment, we provided information about Italy in the introduction.
- I believe that describing the paper as a “systematic review” is not completely appropriate, since systematic reviews are usually meant to follow specific guidelines (e.g., PRISMA) concerning the search strategy and the paper presentation. The term “systematic review” may sound confusing in this case.
- We have simplified this passage, but we used the same systematic review methodology according to PRISMA. In this sense it is appropriate to talk about systematic review search. Thanks to this comment, we converted the PRISMA model created by us into the current PRISMA diagram.
- In the Discussion section, I would suggest to add some sentences concerning the possible clinical implications of the results from this search. How would these be useful for improving the clinical management of bipolar disorders?
- This is a very important aspect; thank you for giving us the opportunity to better improve the discussion section - we added a specific paragraph.
- The Limitations section should be expanded a bit. Indeed, there are some further points that could be addressed, e.g., some of the inclusion criteria are merely operational, such as the choice to include only the first 20 websites in each search engine.
- Thank you so much. We selected the first 20 websites in line with the literature that show how people rarely go beyond the first 2 pages (Eysenbach et al., 2002; Marwaha et al., 2013). For the specific example, we have explained this in the methods paragraph. If there are other limits that we did not declare, we will be pleased to provide further explanation.
- There is something wrong with references cited along the text since they do not appear in consecutive order.
- Thank you for having noticed that. We corrected the order.
- Punctuation and spaces should be revised. English language in some sentences should be revised as well.
- Thank you for having noticed that. We revised it.

Reviewer 2 Report
Dear Editor,
I really appreciate the opportunity to review the manuscript jcm-1861371 entitled:
"The current quality of web-based information on the treatment of the bipolar disorder: a systematic search"
I commend the authors for describing this critical and timely issue. The paper is interesting and well-written; however, I would like to highlight some issues that merit revision:
- I have noted some typos in the text, e.g. "Duck Go"(Page 2, methods; Correct: "DuckDuckGo"), and so on; please, correct
- From the manuscript, which is very interesting and based on well-conducted research work, it is not totally clear to the reader what is meant by "treatment." By this, I mean that the message is clear to an insider, but in times where the press, on necessarily specialized, starts from an article to build the above thesis, I think it is better to specify. I would ask the authors, please, to specify whether the technologies have made it possible to identify pharmacological treatments, psychotherapeutic treatments, or both, possibly commenting in brevity to provide an all-inclusive figure for those seeking information on the topic at hand by any of the means considered.
Author Response
Reviewer #2:
Thank you very much for your kind consideration of our study.
- I have noted some typos in the text, e.g. "Duck Go"(Page 2, methods; Correct: "DuckDuckGo"), and so on; please, correct
- Thank you for having noticed that. We corrected it.
- From the manuscript, which is very interesting and based on well-conducted research work, it is not totally clear to the reader what is meant by "treatment." By this, I mean that the message is clear to an insider, but in times where the press, on necessarily specialized, starts from an article to build the above thesis, I think it is better to specify. I would ask the authors, please, to specify whether the technologies have made it possible to identify pharmacological treatments, psychotherapeutic treatments, or both, possibly commenting in brevity to provide an all-inclusive figure for those seeking information on the topic at hand by any of the means considered.
- Thank you so much for the comment. We have modified the introduction adding more words to explain the different treatments in line with the guidelines for Bipolar disorder. And in the results section as well, we added a paragraph in which we examinate the explanation of different treatments comparing the different websites with reference to the guidelines.
